# An Adenovirus Vector Expressing FMDV RNA Polymerase Combined with a Chimeric VLP Harboring a Neutralizing Epitope as a Prime Boost Strategy to Induce FMDV-Specific Humoral and Cellular Responses

**DOI:** 10.3390/ph14070675

**Published:** 2021-07-15

**Authors:** Giselle Rangel, Verónica Martín, Juan Bárcena, Esther Blanco, Alí Alejo

**Affiliations:** 1Centro de Investigación en Sanidad Animal (INIA, CSIC), Ctra de Algete a El Casar de Talamanca, Valdeolmos, 28130 Madrid, Spain; grangel@indicasat.org.pa (G.R.); veronica.martin@inia.es (V.M.); barcena@inia.es (J.B.); blanco@inia.es (E.B.); 2Instituto de Investigaciones Científicas y Servicios de Alta Tecnología (INDICA-SAT-AIP), City of Knowledge, Panama 0843-01103, Panama

**Keywords:** foot and mouth disease virus, virus-like particle, adenovirus vaccine, prime-boost vaccination

## Abstract

Foot and mouth disease is a highly contagious disease affecting cattle, sheep, and swine among other cloven-hoofed animals that imposes serious economic burden by its direct effects on farm productivity as well as on commerce of farmed produce. Vaccination using inactivated viral strains of the different serotypes is an effective protective measure, but has several drawbacks including a lack of cross protection and the perils associated with the large-scale growth of infectious virus. We have previously developed chimeric virus-like particles (VLPs) bearing an FMDV epitope which induced strong specific humoral responses in vaccinated pigs but conferred only partial protection against homologous challenge. While this and other FMD vaccines under development mostly rely on the induction of neutralizing responses, it is thought that induction of specific T-cell responses might improve both cross protective efficacy as well as duration of immunity. Therefore, we here describe the development of a recombinant adenovirus expressing the highly conserved nonstructural FMDV 3D protein as well as its capacity to induce specific T-cell responses in a murine model. We further describe the generation of an FMDV serotype C-specific chimeric VLP and analyze the immunogenicity of two different prime-boost strategies combining both elements in mice. This combination can effectively induce both humoral and cellular FMDV-specific responses eliciting high titers of ELISA and neutralizing antibodies anti-FMDV as well as a high frequency of IFNγ-secreting cells. These results provide the basis for further testing of this anti FMD vaccination strategy in cattle or pig, two of the most relevant natural host of this pathogen.

## 1. Introduction

Foot and mouth disease (FMD) affects swine, cattle, sheep, and other farmed and wild cloven-hoofed mammals. The disease is characterized by the appearance of a variable number of vesicular lesions in and around the mouth and on the feet [1,2]. While mortality rates are generally low, the productivity in farmed animals is greatly reduced, even over long time periods [3]. Importantly, the disease is highly contagious and can be transmitted by direct contact or over long distances by airborne spread, making its control a great difficulty [4,5]. Overall, FMD poses an important economic burden worldwide both in terms of direct effects of the disease as well as of the effects on the commerce of farmed animal-derived products [6].

The disease is caused by infection with foot and mouth disease virus (FMDV), a prototypical Aphthovirus in the Picornaviridae family [7]. The virus has a small, naked icosahedral capsid with a diameter of about 30 nm that is formed by four distinct structural proteins. Its single stranded, positive polarity RNA genome of about 8.4 kb encodes a single long polyprotein that gives rise, by proteolytic processing, to the distinct structural and non-structural proteins during the infection cycle [8].

Protection against the disease is provided by conventional vaccines composed of inactivated virus and is accompanied by the development of specific neutralizing antibodies. Seven different non-cross protective serotypes of FMDV have been described, namely serotypes A, C, O SAT1, SAT2, SAT3, and Asia 1 with multiple topotypes and variants. These currently circulate in seven independent geographically restricted pools of defined composition requiring specifically tailored diagnostic and prevention measures [9].

Important efforts have been devoted to the development of safer and broadly neutralizing vaccines in the recent years including the generation of attenuated viruses, subunit vaccines, empty stabilized virus-like particles (VLPs), and others. Recently, we have employed modified VLPs to induce specific neutralizing antibody responses as well as partial protection against FMDV serotype-O-caused disease in swine [10]. To this end, we used the well-characterized VP60 protein from rabbit hemorrhagic disease virus (RHDV) which can self-assemble into highly stable approximately 40 nm diameter VLPs when expressed from recombinant baculoviruses [11]. These particles were found to be highly immunogenic and admit the insertion of foreign epitopes at different locations, toward which specific antibodies would be generated in vivo [12]. Structure function analyses have identified exposed sites on the surface of the RHDV VLPs which are amenable to epitope insertion and efficient presentation [13]. A set of modified recombinant engineered RHDV VP60 proteins were expressed to assemble into bona fide chimeric VLPs bearing an exposed FMDV serotype O derived epitope corresponding to the GH loop of the VP1 protein on their surface [14]. These particles were found to induce high FMDV-neutralizing antibody titers both in a murine and a porcine model. However, failure to induce complete protection against a homologous challenge in swine was attributed in part to an observed lack of specific T-cell responses, even when chimeric VLPs bearing an FMDV T-cell epitope were used [10].

While the contribution of T-cell responses toward protection against FMDV by vaccination have not been clearly established yet, it has been proposed to be a potential factor in the generation of long-lasting and cross-serotype protection. Additionally, heterologous prime-boost immunization protocols have proven to be successful approaches for protection against different diseases and have been used before to provide efficient protection against FMD in cattle and pigs [15,16]. Therefore, we decided to devise a strategy to allow the consistent induction of FMDV-specific T-cell responses to complement the good humoral response induced by our previously devised chimeric VLPs in a potentially more efficacious prime boost vaccination regime. Various recombinant adenoviruses expressing structural proteins have been successfully used in the prevention of FMD of different serotypes in cattle and swine and one such virus is the sole novel generation vaccine to have been authorized for emergency use by the U.S. Department of Agriculture (USDA) [17]. Here we describe the generation of a recombinant adenovirus expressing the non-structural, highly conserved FMDV 3D protein and describe its ability to induce specific cellular responses in vivo. Using a murine model, we further analyze two different prime boost regimes of this vector in combination with newly developed, serotype C-specific and neutralizing antibody-inducing chimeric VLPs.

## 2. Results and Discussion

### 2.1. Generation of Recombinant Adenoviruses Expressing the FMDV 3D RNA Polymerase

The region encoding the FMDV 3D RNA-dependent RNA polymerase mediating viral transcription and replication has been shown to be better conserved than other regions encoding structural proteins [18]. Moreover, the 3D protein contains specific T-cell epitopes recognized by lymphocytes from infected swine [19,20] and cattle [21] and a recombinant vaccinia virus expressing it was shown to induce specific T-cell responses and confer partial protection upon FMDV challenge in swine [19]. Further, human adenovirus type 5-based vectors have been shown to be safe and potent inducers of both humoral and cellular responses in different farmed animal species and one such virus [22] has received emergency use authorization for the prevention of FMD in swine. Therefore, we decided to generate a recombinant defective human adenovirus 5-based vector to express the full-length 3D RNA polymerase from FMDV. To avoid potential interference of the viral RNA polymerase during generation of the recombinant virus, a second adenovirus bearing a mutant 3D RNA polymerase (K18A, K20A) with reduced catalytic activity [23] was also generated as a backup strategy. Briefly, protein 3D- and protein 3DKK-encoding genes were amplified by PCR and directly recombined into human adenovirus type 5-based defective genomic vectors under the control of the strong eukaryotic CMV promoter. The linearized recombinant adenoviral DNAs were transfected into HEK293 producer cells. Infected cells were recognized by expression of the marker green fluorescent protein under a fluorescence microscope and initial viral seeds recovered at 72–96 h post-transfection. Viral stocks were then generated by two consecutive rounds of amplification on HEK293 cells followed by a density gradient, ultracentrifugation-based purification protocol. These highly purified viral stocks of rhAdV5-3D and rhAdV5-3DKK were titrated by direct fluorescence detection and used for all subsequent experiments. As a control, a recombinant adenovirus bearing the lacZ reporter gene was generated following the same approach.

We next infected non-complementing Vero cell cultures with the recombinant adenoviruses rhAdV5-3D and rhAdV5-3DKK or rhAdV5-lacZ and observed gfp expression under the fluorescence microscope in all of them (Figure 1A), as expected. Analysis of samples of similarly infected cells by Western blot using an anti FMDV 3D RNA polymerase mAb showed a band of the expected size in the corresponding samples that was absent from uninfected or rhAdV5-lacZ-infected samples and coincided in mobility with that of the purified recombinant protein (Figure 1B). This confirms that the generated adenoviruses are able to express the FMDV 3D polymerase in cell culture.

### 2.2. Recombinant Adenoviruses Expressing the FMDV 3D RNA Polymerase Can Induce Specific Cellular Responses In Vivo

To address whether the recombinant adenoviruses could induce specific T-cell responses in vivo, we inoculated groups of ten C57BL/6 mice subcutaneously with 10^9^ IFU/animal of either rhAdV5-3D, rhAdV5-3DKK, or rhAdV5-lacZ or with PBS in the control group. Two inoculations at days 0 and 14 were performed and serum and spleen samples from 5 animals in each group were collected at day 21, corresponding to day 7 post-boost. At this time point, no detectable anti FMDV 3D polymerase antibodies were found in any of the samples (not shown), showing that the expressed proteins did not induce a good antibody response under these experimental conditions. The non-structural FMDV 3D protein is known to induce antibody responses in both infected as well as vaccinated animals [24,25]. The lack of response to the 3D protein in this case may reflect the competition with the strong antibody response known to be induced in mice against both the adenovirus hexon [26] and fiber proteins [27], although a poor presentation or expression level cannot be excluded. In concordance with this result, no neutralizing activity against FMDV was detected in any of the tested sera either (not shown). We used ELISPOT assays to detect specifically activated cells producing IFNγ in response to incubation with recombinant 3D protein in splenocytes from all experimental groups. As shown in Figure 2A, animals inoculated with rhAd5-3D showed a significant increase of responding cells as compared to either rhAd5-lacZ-inoculated animals or PBS-inoculated animals from the control group. Animals inoculated with Ad-3DKK also showed higher mean numbers of IFNγ-expressing cells as the control groups, although the differences in this case were not significant. Altogether, this experiment shows that expression of the FMDV 3D polymerase from a recombinant adenovirus can elicit an FMDV-specific cellular response in vivo. The C57BL/6 mice have been shown to be susceptible to FMDVC-S8c1 infection, to which they rapidly succumb and from which they can be efficiently protected by vaccination with a conventional inactivated FMDV vaccine [28]. Therefore, we next challenged the remaining 5 mice per group at day 23 (day 9 post-boost) with a lethal dose of FMDVC-S8c1. No protection in terms of mortality was observed, with all animals from the four groups succumbing to infection by day 2 post-challenge (Figure 2B). This shows that the T-cell response elicited by Ad-3D or Ad-3DKK inoculation alone was not sufficient to protect the infected mice.

This result is consistent with previous studies in mice, in which full protection only occurred in the presence of neutralizing antibodies. This is the case, for example, for a Vaccinia virus-based FMDV vaccine expressing the complete polyprotein precursor of the FMDV structural proteins VP1-VP4 [29]. A vaccine based on a defective FMDV population that was similarly found to protect mice from infection did induce a specific T-cell response as well as neutralizing antibodies [30]. The importance of T-cell responses for protection against FMDV in relevant host species has not been extensively studied. However, in cattle specific CD4^+^ T-cell responses are thought to be essential for the development of neutralizing antibody during vaccination [31] and to correlate with the degree of protection conferred [32]. Additionally, FMDV specific CD8^+^ T-cell responses have been detected in both vaccinated and naturally infected animals against peptides that are shared among different viral serotypes [33,34]. The induction of specific T-cell responses might be of relevance for the development of sustained and fully protective responses against FMDV infection as well as for the generation of heterotypic protection. As mentioned above, vaccination of swine with recombinant adenoviral vectors bearing an FMDV non-processable polyprotein induced a T-cell response with specific anti-FMDV CTL-killing activity and low neutralizing antibodies [35]. While this vaccine did not provide full protection from FMDV challenge, it did reduce viremia significantly [36] suggesting that an effective T-cell response may contribute to vaccine-provided protection. Overall, the previous evidence supports the notion that FMDV vaccines inducing T-cell responses such as the one provided by Ad-3D may be used in concert with neutralizing antibody-inducing agents to improve the level or duration of the protection provided.

### 2.3. Generation of Chimeric VLPs Bearing an FMDV Neutralizing Epitope from Serotype C

We recently reported that a chimeric RHDV VLP vaccine candidate containing the FMDV B-cell epitope from O UK/01 serotype as immunogen induces a potent humoral response both in murine and porcine models. Further, vaccination of pigs induced high titers of neutralizing antibodies and conferred partial protection upon homologous challenge [10].

Because no cross neutralization among FMDV serotypes is observed and the employed murine model has not been reported to be susceptible to serotype O FMDV infection to our knowledge, we next generated potentially equivalent chimeric VLPs bearing the equivalent B epitope derived from the GH loop of protein VP1 from FMDV_CS8c1 isolate. This epitope corresponds to a hypervariable major antigenic site in FMDV and has been variously used to induce specific neutralizing antibodies before. This sequence was inserted between nucleotides coding for residues 306 and 307 of the VP60 protein, corresponding to a loop in the exposed P2 structural subdomain [37] and the resulting construct named VP60_CS8c1 (Figure 3). As a control, baculoviruses expressing VP60 without foreign epitope or with the equivalent epitope from FMDV serotype O (VP60_OUK) were used in parallel. The recombinant proteins were purified following previously established VLP obtention protocols [10] and Western blot analyses using anti-VP60 and anti FMDV_CS8c1 peptide (SD6) antibodies confirmed the presence of the expected epitope sequence in the chimeric particle (Figure 3A). The purified particles were then analyzed by ELISA using the SD6 antibody to confirm the presence of the newly inserted peptide sequence (Figure 3B). Additionally, the monoclonal antibody 2E7, that recognizes a lineal epitope at the C-terminal end of the VP60 protein [38] detected all three constructs equally, indicating similar quantities of VLPs were being analyzed. However, the signal provided by antibody 1C9 relative to that detected by 2E7 was considerably lower in both VP60_OUK and VP60_FMDVCS8c1 as in VP60. Because 1C9 is directed against a conformational epitope affected by insertion of chimeric sequences into the exposed VP60 loop [38], we interpret this result to show that insertion of both the control and the relevant epitopes occurred as expected. Altogether, these results extend previous reports [10,13] to show insertion of a new antigenic peptide at the exposed loop of RHDV VP60 for the generation of chimeric VLPs and confirms the FMDV serotype C specificity of these novel particles.

### 2.4. A Combined Vaccination Scheme Including rhAdV5-3D and RHDV-FMDV C_S8c1 VLPs Induces Both Specific Neutralizing Antibodies and Cellular Responses in Mice

We next performed a vaccination assay using the murine C57BL/6 model as before and two different schemes in which animals were primed at day 0 with either VLP or adenovirus inoculation and then boosted with two consecutive inoculations of the converse agent. In schedule 1, animals were vaccinated at day 0 and day 21 with VLPs and at day 28 with adenovirus. In scheme 2, adenovirus was inoculated at day 0 and VLPs at days 8 and 28. Samples were taken on day 35 in both cases from groups of five animals, sparing the remaining five for a challenge experiment (Figure 4A). Sera were obtained at days 21 and 35 of the experiment and the presence of antibodies against purified VP60 protein and the FMDV-CS8c1 peptide analyzed by ELISA. All groups that had been exposed to VLPs, but not the control PBS inoculated group had developed high anti VP60 antibody titers by day 35 of the experiment (Figure 4B). Moreover, all the groups that had received VP60_FMDVCS8c1, but not those vaccinated with either wild type VLP (VP60) or the VLP bearing the epitope from the FMDV O serotype (VP60_OUK) developed specific antibodies against the FMDV-CS8c1 peptide (Figure 4C). This confirms the immunogenicity of RHDV-based VLPs and expands the use of this proposed insertion site to further virus-derived epitopes, supporting the proposed versatility of this platform as a tool for vaccine development. The specific antibody titers increased after a boost with the corresponding VLPs. Relevantly, at day 35, all sera from the VP60_FMDVCS8c1 vaccinated groups showed specific FMDV CS8c1-neutralizing activity (Figure 4D). The neutralizing titers were elevated and not significantly altered by the order of inoculation, showing that vaccination with recombinant adenovirus did not impair generation of neutralizing antibodies.

Finally, splenocytes were obtained at day 35 and used to detect specific cellular responses directed against FMDV 3D protein as above. In this case, only the groups exposed to rhAdV5-3D, but not those that received the control rhAdV5-lacZ virus developed a significant response (Figure 5A). The magnitude of the response was similar in the case of both vaccination schemes showing on the one hand that prior inoculation with VLPs did not affect T-cell response induction by the adenoviruses and on the other hand that these responses were initiated as soon as 7 days post-inoculation and maintained at least up to 35 days post-inoculation. Finally, the remaining mice were challenged with 1 LD50 equivalent dose of FMDVC-S8c1 to analyze protection. While animals vaccinated with the VLP bearing the epitope from the non-crossreacting FMDV serotype O succumbed to the infection to the same degree as the control group, vaccination with VLP VP60_FMDVCS8c1 conferred full protection in terms of mortality independently of the vaccination scheme and of the presence of rhAdV5-3D (Figure 5B).

This shows that the newly developed chimeric VLP can induce the generation of neutralizing antibody responses and confer protection against disease caused by infection with the homologous FMDVC-S8c1 in the murine model. Experiments using higher doses of challenge virus will be required to allow an assessment both of the potency and duration of chimeric VLP vaccination on its own as well as of the effect of adenovirus-mediated induction of FMDV-specific T-cell responses on protection. Further, the more relevant porcine model may be used to characterize both the T-cell response induced by the newly developed adenovirus as well as its effect on the efficacy in protection in combination with neutralizing antibody-inducing chimeric VLPs.

## 3. Materials and Methods

### 3.1. Cells and Viruses

HEK293 (ATCC CRL 3216), BHK-21 (ATCC CCL10), and Vero (ATCC CCL81) cells were grown in DMEM supplemented with 5% FCS at 37 °C in a 5% CO_2_ humidified atmosphere. The insect cell line Hi5 [39] was grown in TC100 medium supplemented with 10% FCS at 28 °C. Murine splenocytes were obtained by mechanical disaggregation followed by filtration through 70-µm pores in RPMI 1640 (BioWhittaker) medium supplemented with 5% FCS. Recombinant adenovirus stocks were amplified on HEK293 monolayers and purified following a CsCl density gradient ultracentrifugation procedure described in [40]. Titration of viral stocks was performed by direct fluorescence detection on HEK293 monolayers and titers consistently range between 1 × 10^11^ pfu/mL and 1 × 10^12^ pfu/mL. Recombinant baculovirus stocks were amplified on Hi5 cells and maintained at 4 °C. FMDV_CS8c1 stocks were grown and titrated on BHK-21 cells as described [41].

### 3.2. Generation of Recombinant Adenoviruses and Baculoviruses

The full-length gene encoding FMDV RNA polymerase 3D or the K18A, K20A mutant (3DKK) [23] were amplified by PCR from plasmids encoding each of the constructs which were kindly donated by the laboratory of Dr. E. Domingo (Centro de Biología Molecular “Severo Ochoa”, Spain). These PCR products were cloned by InFusion cloning into prelinearized pAdenoX-ZsGreen 1 vector (Adenox Adnoviral System 3, Clontech) following the manufacturer’s instructions. The correctness of the insert was verified by Sanger sequencing before transfecting the obtained adenovirus vectors into HEK293 cells. Initial recombinant rhAdV5-3D and rhAdV5-3DKK virus stocks were harvested at 72 h post-transfection and amplified following the provided guidelines. These viruses carry the foreign genes under the control of the strong eukaryotic CMV promoter as well as a green fluorescent reporter to monitor transduction and infection efficiency. To obtain the recombinant baculovirus to express the chimeric RHDV-based VLPs, the sequence encoding the desired FMDV peptide was inserted by Q5 site directed mutagenesis (NEB) into the pBac1-based plasmid pAH109 [38] that bears the RHDV VP60-encoding gene under control of the late baculoviral polyhedrin. The peptide sequence (TTTYTASARGDLAHLTTTHARHLP) includes the G-H loop of the FMDV-CS8c1 VP1 protein S8c1 [42] flanked by two GS residues on both its N- and C-terminal ends to allow flexible insertion. The corresponding oligonucleotides were designed using the NEB online tool NEBasechanger (http://nebasechanger.neb.com, accessed on 7 July 2017) to allow in frame insertion of the foreign sequence between sites encoding residues 306 and 307 of the VP60 protein. The absence of unwanted mutations in the construct was confirmed by Sanger sequencing. Recombinant baculoviruses were obtained in a single step by co-transfection of the obtained plasmids with the linearized baculoviral DNA BACUltra (Oxford Expression Technologies) and amplified and stored as recommended by the manufacturer. Details of cloning strategy and final sequence of the full-length plasmid obtained are available upon request. The chimeric FMDV serotype O VLP was generated in the same way and has been described before [10]. For reference, the latter bears the equivalent peptide (PVTNVRGDLQVLAQKAART) from the VP1 protein from FMDV_O_UKG/35/2001 [43].

### 3.3. Purification of Virus-Like Particles

The recombinant baculoviruses expressing RHDV VP60 or the chimeric versions including the VP1 G-H loop-derived major antigenic site peptide from FMDV_CS8c1 (VP60_CS8c1) or FMDV_OUKG/35/2001 (VP60_OUK) were used to infect Hi5 cell monolayers at a high multiplicity of infection (MOI) (>10 pfu/cell) and the cells were harvested at 72 hpi, when the cytopathic effect was evident. VLPs were obtained as described before [10] by solubilization in the presence of detergent, low-speed clarification and ultracentrifugation through a sucrose cushion. Finally, particles were concentrated by ultracentrifugation and resuspended in 0.2 M sodium phosphate, 0.1 M NaCl, pH 6.0 buffer for storage at 4 °C. The purified proteins were analyzed by SDS-PAGE and quantified by a BCA assay (Pierce).

### 3.4. Protein Transfer and Immunodetection on Membranes (Western Blot)

Protein samples disrupted in Laemmli buffer were subjected to denaturing gel electrophoresis and subsequently transferred to PVDF membranes. Membranes were blocked with PBS supplemented with 5% (*w*/*v*) BSA for 1 h and then incubated for 16 h at 4 °C with the antibodies 2E7 and 1C9 [38], recognizing the RHDV VP60 protein, the mAb SD6, that recognizes the neutralizing epitope found in the G-H loop of protein VP1 from FMDV_CS8c1 [44], the mAb 3F12 [45], that recognizes FMDV 3D protein (residues 29–38) or a commercial anti-GAPDH mouse monoclonal (mAbcam 9484), as a loading control. After several washes the membranes were incubated for 1 h with HRP-conjugated secondary antibodies as required and developed using PierceTM ECL Plus Western Blotting Reagent (Thermo Fisher Scientific) following commercial recommendations.

### 3.5. Immunodetection by ELISA Assays

Sera from immunized mice were analyzed by ELISA for the presence of antibodies against purified RHDV VP60 protein or FMDV_Cs8c1 B peptide as described before [10], as well as against FMDV non-structural protein 3D. The FMDV_Cs8c1 B peptide was synthesized in the laboratory of D. Andreu (University Pompeu Fabra, Barcelona, Spain) and the recombinant purified FMDV 3D protein was a kind gift from the Esteban Domingo lab. Peptide (1 µg/well), recombinant purified RHDV VP60 (300 ng/well), or FMDV 3D protein (100 ng/well) were bound to 96-well high binding plates (Corning) for the peptide or Polysorp 96-well ELISA plates (Nunc) by overnight incubation at 4 °C in carbonate-bicarbonate buffer. Bound murine antibodies were detected by HRP-conjugated goat anti-mouse IgG (Invitrogen) followed by OPD or TMB detection, respectively. Reaction was stopped by the addition of 3 or 1.8 N H_2_SO_4_ and absorbance was read at 492 or 450 nm on a Fluostar Omega microplate reader. For analyses of VLPs, 200 ng per well of the corresponding purified particles was bound and detection was performed as above using mAbs 2E7, 1C9, or mAbSD6.

### 3.6. Detection of Specific IFNγ-Secreting Cells by ELISPOT

ELISPOT IFNγ assays were performed as described before [10]. Briefly, splenocytes were stimulated with 2.5 µg/mL of purified recombinant 3D protein (kindly donated by the E. Domingo laboratory) for 24 h and transferred to monoclonal anti mouse IFNγ-capture antibody-coated Multiscreen HA Filter 96-well plates (Merck). Cytokine-producing cells were detected with the biotynilated anti -IFNγ detection antibody (Becton Dickinson) followed by HRP-Streptavidin and 3-amino-9-ethylcarbazole (AEC). Positive cells were counted under a magnifying glass and are represented as a mean of triplicate values for each individual analyzed.

### 3.7. FMDV Neutralization Assay

FMDVCS8c1-specific neutralizing antibodies in sera were quantified using a microtritration assay on BHK21 cells as described before and are expressed as the extrapolated reciprocal number of the highest dilution of serum capable of reducing by 50% the number of infected wells by a 100 TCID_50_ dose of the virus.

### 3.8. Immunization and Challenge of C57BL/6 Mice

Groups of five 6–8 weeks old female C57BL/6 mice (Harlan) were housed in individually ventilated cages within a biocontainment level 3 laboratory. Mice were inoculated subcutaneously with 100 µg of purified VLPs emulsified in a 1:1 (*v*/*v*) ratio with the Montanide ISA 50V2 adjuvant in a final volume of 100 µL at 21 day intervals. Recombinant adenoviruses were injected subcutaneously at a dose of 10^9^ IFU per animal in a 100 µL volume. Blood samples were obtained from the submandibular vein at the indicated time points. Challenge was performed by intraperitoneal inoculation of 10^3^ TCID_50_ of FMDV_CS8c1 per animal or with a sublethal dose of 10^2^ TCID_50_ per animal for the prime boost experiment. All animals were monitored daily for general health status and handled according to the current regulations of the EU and Spanish Government. The experimental design and procedures employed were approved by the Ethics Committee (CEEA2013/009) and Biosecurity committee (CBS 2013/006) from INIA under reference PROEX 228/14, approved on 24 November 2014.

## 4. Conclusions

In summary, we have described the generation of a novel chimeric VLP that provides protection against disease caused by FMDC-S8c1 infection in vivo. Additionally, we provide functional novel adenovirus vectors to induce cellular responses against the FMDV 3D protein which is highly conserved among serotypes and may therefore help in the establishment of both heterotypic as well as long-lasting protective responses against FMD.

## Figures and Tables

**Figure 1 pharmaceuticals-14-00675-f001:**
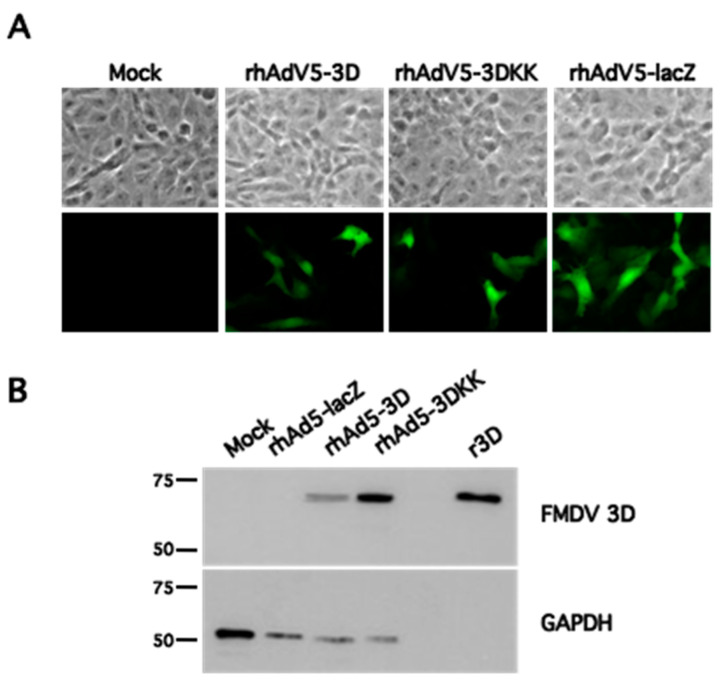
Generation of recombinant adenoviruses expressing the FMDV 3D RNA polymerase. (**A**), Vero cells were mock infected or infected with the indicated recombinant adenoviruses at an MOI of 1 IFU/cell and fixed at 24 hpi. Upper panel shows phase contrast image of selected fields and lower panel shows GFP expression as analyzed by fluorescence microscopy. (**B**), Vero cells were mock infected or infected as indicated at an MOI of 10 IFU/cell and harvested at 24 hpi. Samples were analyzed by Western blot using antibodies against FMDV 3D protein or GAPDH as shown. A control lane containing 10 ng of purified recombinant FMDV 3D protein (r3D) was included as a reference and specificity control. The position of MWM (kDa) is indicated on the left.

**Figure 2 pharmaceuticals-14-00675-f002:**
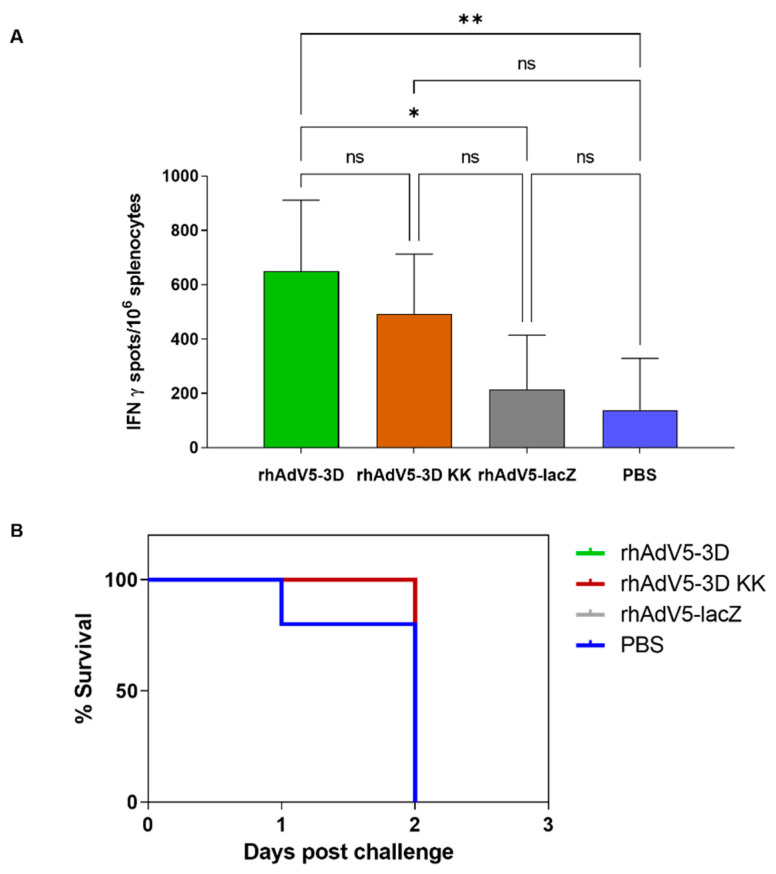
The recombinant adenoviruses expressing FMDV 3D protein induce a specific cellular response in vivo. In (**A**), ELISPOT analysis of IFNγ producing splenocytes in response to stimulation with recombinant protein FMDV 3D from groups of five C57BL/6 mice inoculated subcutaneously with 10^9^ IFU of the indicated adenoviruses at days 0 and 14 of the experiment and harvested at day 21. Shown are means + SD for each group of 5 mice and significant differences among groups indicated by ** (*p* < 0.01) or * (*p* < 0.05), ns: no significant differences. In (**B**), groups of five C57BL/6 mice vaccinated as above were challenged via intraperitoneal inoculation at day 23 of the experiment with FMDVC-S8c1 and survival recorded.

**Figure 3 pharmaceuticals-14-00675-f003:**
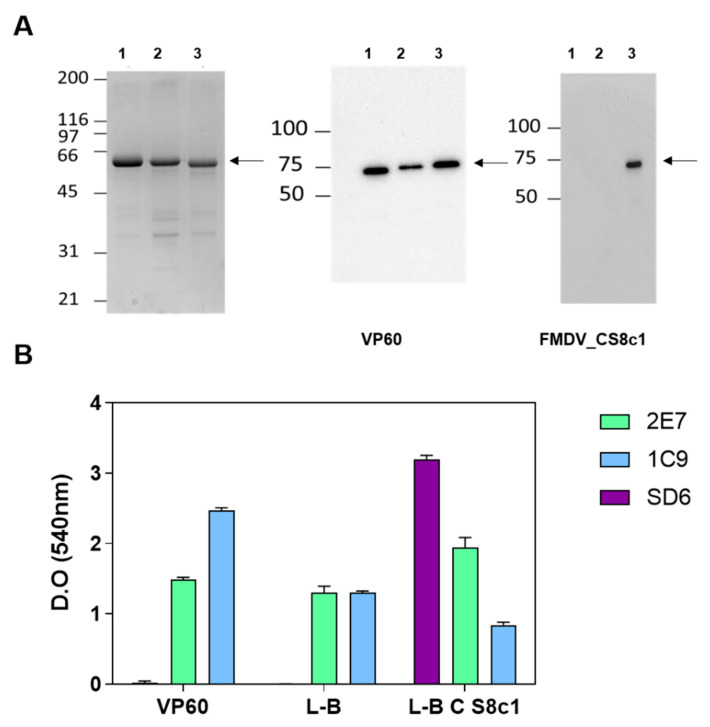
Generation of RHDV VP60-based chimeric VLPs bearing an FMDV_CS8c1 B epitope. In (**A**), purified recombinant VP60 (1), VP60_OUK bearing the FMDV_OUK isolate epitope (2) or VP60_CS8c1, bearing the FMDV_CS8c1 non-crossreacting epitope (3) were analyzed by Coomassie blue staining (left panel) or Western blot (middle and right panels) with the indicated antibodies. Arrows show position of detected proteins and MWM (kDa) are indicated on the left. In (**B**), samples from the indicated purified proteins were analyzed by ELISA using the antibodies shown in the legend: 2E7 and 1C9 mAbs recognize RHDV VP60 epitopes, with 1C9 corresponding to a conformational epitope around the insertion site used in this approach; SD6 mAb recognizes the neutralizing epitope found in the G-H loop of protein VP1 from FMDV_CS8c1. Shown are mean + SD of triplicates from one representative assay out of two.

**Figure 4 pharmaceuticals-14-00675-f004:**
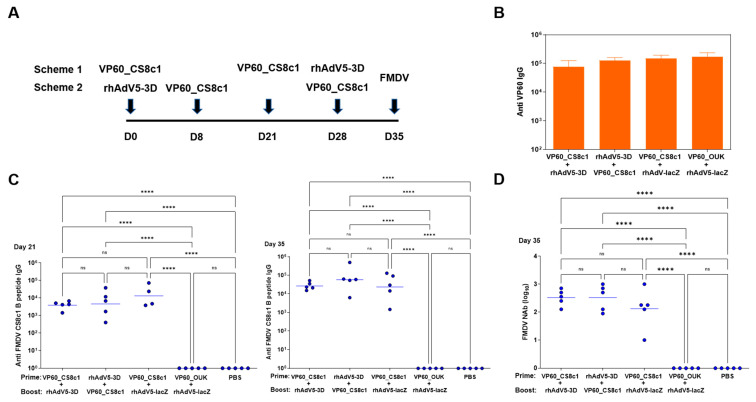
Specific humoral anti-FMDV responses in VLP/Adenovirus prime boost vaccination regimes. In (**A**), a schematic view of both regimes used is shown. Groups of ten C57BL/6 mice were vaccinated using sequential adenovirus and VLP combinations. In scheme 1, animals were vaccinated at day 0 and day 21 with 100 µg of purified VP60_CS8c1 VLP and at day 28 with a single subcutaneous inoculation of 10^9^ IFU of rhAdV5-3D. In scheme 2, rhAdV5-3D was inoculated at day 0, and VP60_CS8c1 at days 8 and 28. Samples were taken on day 35 in both cases from groups of five animals, sparing the remaining five for the challenge experiment. Corresponding control groups using either rhAd5-lacZ or VP60_OUK VLP or PBS-inoculated animals were included. IgG anti-VP60 antibody titers in sera at day 35 (**B**) or anti FMDV_CS8c1 B-epitope peptide titers at days 21 and 35 (**C**) for the indicated groups as determined by ELISA. In (**B**) mean + SD for each group is shown. In (**C**), data from individual samples assayed in triplicate is shown and horizontal bars indicate mean titer for each group. In (**D**), FMDV_CS8c1-specific mean-neutralizing antibody titers were calculated by microneutralization assays on BHK-21 cells for each animal in the group and mean titer is indicated by horizontal bars. Significant differences among groups are indicated by **** (*p* < 0.0001), ns: no significant differences.

**Figure 5 pharmaceuticals-14-00675-f005:**
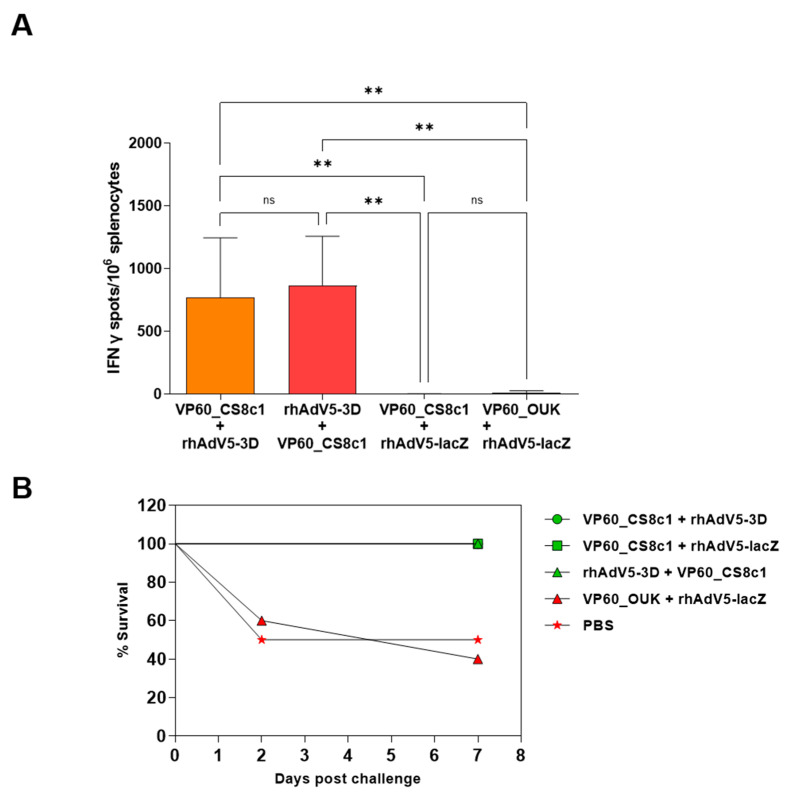
Cellular response and protection upon FMDV challenge in VLP/Adenovirus prime boost vaccination regimes. (**A**) ELISPOT analysis of IFNγ-producing splenocytes in response to stimulation with recombinant protein FMDV 3D. Shown are means + SD for each group of 5 mice as indicated and significant differences among groups indicated by ** (*p* < 0.01), ns: no significant differences. (**B**) Groups of mice as shown where challenged with FMDV_CS8c1 by intraperitoneal inoculation upon completion of the vaccination schedule and survival over an 8-day period is represented.

## Data Availability

Data is contained within the article.

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
