# Peer review of "An Adenovirus Vector Expressing FMDV RNA Polymerase Combined with a Chimeric VLP Harboring a Neutralizing Epitope as a Prime Boost Strategy to Induce FMDV-Specific Humoral and Cellular Responses"

_pharmaceuticals, 2021, doi:10.3390/ph14070675_

Round 1
Reviewer 1 Report
General comment:
In this study, the authors have generated adenovirus vector vaccines expressing FMDV 3D protein to elicit cellular immune responses as well as neutralizing antibody inducing VLPs expressing VP1 of representative serotype C. Although the adenovirus vector vaccine induced moderate level of antigen-specific T cell responses, it failed to protect immunized mice from lethal FMDV infection. In addition, a combination strategy using two different vaccines was described for the purpose of induction of both humoral and cellular responses.
Although this study addresses a significant issue in the FMDV vaccine development, the content of the paper provides limited information and some experiments have not been executed properly (see specific comments below).
Specific comments:
- In Figure 1, growth property data need to be included to provide kinetics and peak production of the recombinant adenoviruses expressing target antigens.
- In Figure 2A, have you tried FACS analysis of IFN producing splenocyte? FACS would be better to test CD4+ and CD8+ T cell responses separately so that undetectable level of FMDV 3D-specific antibodies can be explained.
- In Figure 2 and 5, it would be important to incorporate data for viremia and body weight change.
- Throughout figure 4 and 5, the VLP only group (VP60_CS8c1) needs to be included as a control group.
- In order to show advantages of the vaccine combination strategy, additional data are required such as challenge with higher doses of virus (as the authors mentioned in the text) and different serotypes as well.
Author Response
We wish to thank both reviewers for their time in reading and commenting on our manuscript. Wherever possible we have modified the manuscript according to their input and we sincerely hope that it will be acceptable for publication in this revised form. Our point by point response to each of them are as follows:
Point by point response to reviewer 1
- During the adenovirus stock amplification procedure we do not routinely perform growth kinetic assays, although green fluorescence expression is monitored as a surrogate measure of infection and virus production. Cells are harvested when all of the monolayer is confirmed to be positive for gfp expression, which is usually between three and seven days post infection, depending on the amplification cycle and initial multiplicity of infection. Our final virus stocks obtained by density gradient purification range between 1X1011 pfu/ml and 1X1012 pfu/ml, which is in the range of the expected values for this approach. The latter information is included as a single sentence in the methods section of the revised manuscript.
- Unfortunately we did not assess cellular response by FACS analysis on this occasion as we felt ELISPOT assays would be sufficient for this assessment.
- The experiments shown in Figures 2 and 5 were designed as challenge experiments using the murine FMDV infection model. Because virus induced mortality is very rapid in this model (usually occurring between 24 and 72 hours post infection) additional, separate groups to allow for safe test bleeding without affecting the course of disease were not inlcuded and therefore no data for viremia are available. Additionally, as far as we are aware of, FMDV infection does not lead to a detectable weight loss in this model and was therefore not determined. For references see Salguero FJ et al. Foot-and-mouth disease virus (FMDV) causes an acute disease that can be lethal for adult laboratory mice. Virology. 2005 Feb 5;332(1):384-96 or Cacciabue M et al. Differential replication of Foot-and-mouth disease viruses in mice determine lethality. Virology. 2017 Sep;509:195-204. However, we believe that this does not affect our main conclusions as our aim in this case was to determine a direct effect on protection in terms of prevention of lethality.
- The animals in the experimental groups shown throughout Figures 4 and 5 were subjected to consecutive inoculations of recombinant adenoviruses and VLPs. We used as controls equivalent recombinant adenoviruses expressing lacZ (an irrelevant antigen) or bearing the FMDV serotype O peptide (an epitope that does not cross react with FMDV serotype C used in the challenge experiment) because these mimic more precisely the experimental conditions in our view.
- We fully agree with the reviewer on this point. The data in this manuscript show that a recombinant adenovirus expressing FMDV 3D protein induces a specific T cell reposne in a murine model and that this response does not protect from virus induced lethality. Moreover, we show that in combination with specific chimeric VLPs, both neutralizing humoral responses as well as cellular responses can be attained in vivo. Our experiments in progress in the porcine species as well as with different serotypes will hopefully address the issue of the efficacy of such a combined vaccination approach in a more relevant host.
Please note that other minor modifications in the revised manuscript are:
- The reference number for the authorization by the Ethics Committee (CEEA2013/009) and Biosecurity committee (CBS 2013/006) from INIA for animal experimentation has been corrected to PROEX 228/14 both in the Methods sectin and in the Institutional Review Board Statement.
- An additional person has been included in the acknowledgment section.
Reviewer 2 Report
This manuscript describes the immunogenicity and mice challenge results for a novel adenovirus vector expressing the RNA-dependent RNA polymerase from foot and mouth disease virus (FMDV), combined with a chimeric virus-like particle containing an epitope known to induce neutralizing antibody. The purpose of this study was to investigate the protection potential of a recombinant adenovirus-based vaccine for foot and mouth disease in cloven-hoofed mammals. The study design is novel and experimentally sound and provides a potential alternative vaccine platform from the currently used inactivated FMDV. Furthermore, the report is clearly written, and the conclusions are supported by the data presented. My only two comments are editorial in nature. First, in the Introduction, I suggest that the authors change "different non-crossneutralizing serotypes" (line 56) to "different non-cross protective serotypes". Second, delete the "s" in "extends" in line 242.
Author Response
We wish to thank both reviewers for their time in reading and commenting on our manuscript. Wherever possible we have modified the manuscript according to their input and we sincerely hope that it will be acceptable for publication in this revised form.
Point by point response to reviewer 2
The suggested modifications have been included.
Additionally, a Conclusions section has been included as requested. The final concluding paragraph of the Results and Discussion section has been moved to this new location.
Other minor modifications in the revised manuscript are:
- The reference number for the authorization by the Ethics Committee (CEEA2013/009) and Biosecurity committee (CBS 2013/006) from INIA for animal experimentation has been corrected to PROEX 228/14 both in the Methods sectin and in the Institutional Review Board Statement.
- An additional person has been included in the acknowledgment section.
Round 2
Reviewer 1 Report
In terms of novel vaccines, the chimeric VLP generated in this study shows a reasonable level of protection even though additional cellular responses by the rhAdV5-3D do not have clear advantages.